# Effects of Early Intervention with Maternal Fecal Microbiota and Antibiotics on the Gut Microbiota and Metabolite Profiles of Piglets

**DOI:** 10.3390/metabo8040089

**Published:** 2018-12-06

**Authors:** Chunhui Lin, Jiajia Wan, Yong Su, Weiyun Zhu

**Affiliations:** 1Laboratory of Gastrointestinal Microbiology, Jiangsu Key Laboratory of Gastrointestinal Nutrition and Animal Health, College of Animal Science and Technology, Nanjing Agricultural University, Nanjing 210095, China; 2016105048@njau.edu.cn (C.L.); 2016105049@njau.edu.cn (J.W.); zhuweiyun@njau.edu.cn (W.Z.); 2National Center for International Research on Animal Gut Nutrition, Nanjing Agricultural University, Nanjing 210095, China

**Keywords:** early intervention, fecal microbiota transplantation, metabolite profiles, neonatal pig

## Abstract

We investigated the effects of early intervention with maternal fecal microbiota and antibiotics on gut microbiota and the metabolites. Five litters of healthy neonatal piglets (Duroc × Landrace × Yorkshire, nine piglets in each litter) were used. Piglets in each litter were orally treated with saline (CO), amoxicillin treatment (AM), or maternal fecal microbiota transplantation (MFMT) on days 1–6, with three piglets in each treatment. Results were compared to the CO group. MFMT decreased the relative abundances of *Clostridium sensu stricto* and *Parabacteroides* in the colon on day 7, whereas the abundance of *Blautia* increased, and the abundance of *Corynebacterium* in the stomach reduced on day 21. AM reduced the abundance of *Arcanobacterium* in the stomach on day 7 and reduced the abundances of *Streptococcus* and *Lachnoclostridium* in the ileum and colon on day 21, respectively. The metabolite profile indicated that MFMT markedly influenced carbohydrate metabolism and amino acid (AA) metabolism on day 7. On day 21, carbohydrate metabolism and AA metabolism were affected by AM. The results suggest that MFMT and AM discriminatively modulate gastrointestinal microflora and alter the colonic metabolic profiles of piglets and show different effects in the long-term. MFMT showed a location-specific influence on the gastrointestinal microbiota.

## 1. Introduction

The colonization of the intestinal microbiota commences after birth [1]. Soon after a piglet is delivered, the intestinal microflora rapidly converts from primarily facultative anaerobes into a diverse community of anaerobes [2,3]. Researchers have regarded the gut microbiota as a forgotten organ in the host, owing to the microbial capacity of communicating with one another and the host in different ways [4]. The microbiota profoundly impact the physiology, health, and disease of the host [5]. Recent investigations have suggested that the colonization of the newborn gastrointestinal microbiome plays important roles in the later life of humans and animals, demonstrating predictive power in disease [6,7]. Therefore, in the long-term, the establishment of stable microbiota in early life is beneficial to the individual.

The colonization of the intestinal microbiota is affected not only by internal factors, such as delivery mode [8], but also by environmental factors. Maternal fecal microbiota are one of the environmental factors, since piglets are inevitably exposed to sows’ feces from birth. Our previous study showed that the fecal microbial communities of piglets later during lactation were similar to the composition of the sow’s feces microbiota [9], which suggests maternal fecal microbiota may play an important role in the process of gut microbiota colonization in piglets. So far, information on the effect of early intervention with maternal fecal microbiota is limited. Fecal microbiota transplantation (FMT) is a technique in which feces from a healthy individual is transferred into another’s gut [10]. FMT has attracted researcher attention given its ability to restore intestinal flora, and to cure recurrent *Clostridium difficile* infection as well as symptoms associated with inflammatory bowel disease [11], autism spectrum disorder [12], metabolic syndrome [13], or antibiotic-resistant bacteria [14]. FMT was proven to increase the reshaping of mouse intestinal microbiota [15,16], but this has rarely been investigated in neonatal piglets. Therefore, we attempted to increase the amount of microbiota by FMT to study the effect of early intervention with maternal fecal bacteria on the gut microbiota composition and their metabolism in newborn piglets.

Antibiotics are usually used in the diet to prevent diarrhea [17] and promote growth [18] in pig production; amoxicillin is one of these antibiotics. In recent years, studies have investigated the effects of intervention with amoxicillin on gut flora and metabolism in animals. A previous study suggested that in-feed amoxicillin reduced the numbers of gut microbiota diversity in rats [19]. Amoxicillin produced simplified hindgut bacteria communities with decreased counts in mice [20]. Current studies on the effects of antibiotics on gastrointestinal microflora and metabolism have focused on weaned pigs and growing pigs, whereas few studies have been conducted on newborn piglets.

In the present study, we hypothesized that the gastrointestinal microbiome could be strengthened or weakened by feeding maternal fecal microbiota or antibiotics. Therefore, maternal fecal bacteria and amoxicillin were orally provided to newborn piglets to investigate their short- and long-term effects on intestinal microbiota and metabolites.

## 2. Results

### 2.1. Gastrointestinal Microbial Community

As shown in Table 1, maternal fecal microbiota transplantation (MFMT) significantly increased (*p* < 0.05) the abundance-based coverage estimator (ACE) in the stomach on day 7 compared with the control (CO) group. The Chao value in the MFMT group was greater (*p* < 0.05) than in the amoxicillin treatment (AM) group. On day 21, the diversity Simpson indices in the MFMT group significantly decreased (*p* < 0.05) in comparison to the AM group. In the colon, AM significantly increased the Chao value (*p* < 0.05) on day 21.

At the phylum level, the dominant phylum was *Firmicutes* in both the stomach (Figure 1a) and ileum (Figure 1b) in each group on days 7 and 21. However, the microbiota in the colon were predominated by phyla *Bacteroidetes* and *Firmicutes* (Figure 1c). In the ileum, AM significantly decreased (*p* < 0.05) the abundance of *Proteobacteria* compared to the CO group on day 21.

Genus-level analysis showed that the most predominant genus of the three gut segments was *Lactobacillus* on days 7 and 21. As presented in Figure 2a, in the stomach, genus *Corynebacterium* was significantly increased (*p* < 0.05) in relative abundance by MFMT compared to the AM group on day 7, and the reverse occurred (*p* < 0.05) in comparison to the CO group on day 21. Compared to that in the CO group, the abundance of *Arcanobacterium* in the AM group decreased (*p* < 0.05) on day 7. In the ileum, the abundances of *Veillonella* and *Moraxella* in the MFMT group significantly declined (*p* < 0.05) in comparison to that in the AM group on day 7 (Figure 2b). In comparison to the CO group, AM numerically reduced the abundance of *Streptococcus* (*p* < 0.05) on day 21. In the colon, on day 7, MFMT significantly increased (*p* < 0.05) the abundance of *Blautia* and decreased (*p* < 0.05) the abundances of *Clostridium sensu stricto* and *Parabacteroides* in comparison to that in the CO group (Figure 2c). The relative abundance of *Lachnoclostridium* of the AM group was higher (*p* < 0.05) than that in the CO group on day 21. Compared to the AM group, MFMT significantly increased (*p* < 0.05) the abundance of *Desulfovibrio*.

Operational taxonomic unit (OTU) analysis (Appendix A) demonstrated that in the stomach, the AM group showed a lower abundance of OTU2 (member of *Lactobacillus*) and greater abundances of OTU5 (member of *Lactobacillus*) and OTU6 (member of *Lactobacillus mucosae*) (*p* < 0.05) in comparison to the CO groups on day 7. In the ileum, the AM group showed a lower abundance of OTU31 (member of *Lactobacillus vaginalis*) and a greater abundance of OTU7 (member of *Lactobacillus mucosae*) (*p* < 0.05) in comparison to the CO group (Appendix A) on day 7. On day 21, the AM group showed a lower abundance of OTU42 (member of *Lactobacillus*) and a greater abundance of OTU31 (member of *Lactobacillus vaginalis*) (*p* < 0.05) in comparison to the CO group. In the colon of piglets (Appendix A), the MFMT group showed a greater abundance (*p* < 0.05) of OTU41 (member of *Blautia*) and a lower abundance of OTU5 (member of *Bacteroidales S24-7 group*) than that in the CO group on day 7.

We measured the integrated 16S rRNA copied genes of bacteria in the three gut locations (Figure 3). On day 7, the AM group had the lowest number (*p* < 0.05) of total bacteria in the stomach among the three groups (Figure 3a). In the colon, the number of total bacteria in the MFMT and AM groups showed a significant reduction (*p* < 0.05) compared to that in the CO group, and no difference in the total bacteria number in the three gut segments was found among the three groups on day 21 (Figure 3b).

### 2.2. Metabolite Profiles

The gas chromatography–mass spectrometry (GC–MS)-based measurement detected 98 metabolites in the colon. The key compounds were identified with the orthogonal partial least squares discrimination analysis (OPLS–DA) models (Figure 4). A good discrimination of metabolites among the three groups at the ages of 7 and 21 days is presented in Table 2 with standards of fold change >1.5, *p* < 0.1, and variable importance projection (VIP) >1.

In the colon, five of these metabolites (i.e., asparagine, hypoxanthine, uracil, fumaric acid, and lactic acid) were enriched (*p* < 0.05), and nine (i.e., sucrose, 1-monohexadecanoylglycerol, lysine, 1,3-di-tert-butylbenzene, eicosanoic acid, heptanoic acid, 2,4,6-tri-*tert*-butylbenzenethiol, pipecolic acid, and putrescine) were decreased (*p* < 0.05) in the MFMT group in comparison to the CO group. Pathway enrichment analysis indicated that MFMT affected galactose metabolism, starch and sucrose metabolism, pyruvate metabolism, gluconeogenesis, protein biosynthesis, and the tricarboxylic acid (TCA) cycle (Figure 5a). Five of these metabolites (i.e., glucaric acid, sorbitol, asparagine, fructose, and hypoxanthine) were enriched (*p* < 0.05), and two (i.e., sucrose and oxalic acid) were decreased (*p* < 0.05) in pigs by AM in comparison to the CO group. Pathway enrichment analysis indicated that AM affected galactose metabolism, fructose and mannose degradation, starch and sucrose metabolism, and protein biosynthesis (Figure 5b).

On day 21, seven of these metabolites (i.e., uridine, 2-hydroxyglutaric acid, ethanolamine, arginine, beta-alanine, guanine, and urea) were enriched (*p* < 0.05), and three (i.e., sucrose, citric acid, and ornithine) were decreased (*p* < 0.05) in the MFMT group in comparison to the CO group. Pathway enrichment analysis indicated that MFMT affected galactose metabolism, starch and sucrose metabolism, citric acid cycle, phospholipid biosynthesis, and protein biosynthesis (Figure 5c). Nine of these metabolites (i.e., sorbitol-6-phosphate, uridine, fumaric acid, glucose, ribose, xylose, fructose-6-phosphate, galactose-6-phosphate, and mannose-6-phosphate) were enriched (*p* < 0.05), and two (i.e., cholesterol and glycerol) were decreased (*p* < 0.05) in the AM group in comparison to the CO group. Pathway enrichment analysis indicated that AM affected the glucose–alanine cycle, insulin signaling, galactose metabolism, citric acid cycle, steroid biosynthesis, steroidogenesis, pentose phosphate pathway, glycerolipid metabolism, glycolysis, gluconeogenesis, and fructose and mannose degradation (Figure 5d).

## 3. Discussion

A growing body of research has suggested that the early colonization of microflora affects microbial communities and the metabolic processes of gut microorganisms [6,7,21]. For the first time, as we have seen, the current experiment combined high-throughput sequencing and microbiome analysis to study the short- and long-term effects of early intervention with maternal fecal microbiota and antibiotics on the microbial composition and metabolite profiles in the gastrointestinal tract (GIT) of neonatal piglets.

### 3.1. Effects of Early Maternal Fecal Microbiota and Antibiotics Intervention on the Microbiota of the Gastrointestinal Tract

*Firmicutes* was found to be the most abundant phylum across the different segments of the intestine. This finding is similar to that of previous research indicating that *Firmicutes* had an absolute advantage in piglets [22]. Phylum *Firmicutes* is known to be closely associated with animal energy metabolism [23]. Some researchers found that *Parabacteroides* was responsible for infectious diseases, mainly bacteremia [24]. Therefore, the decrease of *Parabacteroides* in the colon on day 7 in this study showed a potential role of MFMT in strengthening intestinal health. Recent studies demonstrated that some *Parabacteroides* species could protect colon and attenuated colitis [25,26]; however, further study is needed to understand the mechanism of reducing *Parabacteroides* by MFMT. It was reported that *Blautia* produced acetate and succinate, provided as important energy sources [27] that were conducive to colonic health. Some *Clostridium* species are involved in disease occurrence. For example, in Scanlan’s comparison of colon cancer patients with normal individuals, the diversity of *Clostridium coccoides* and *Clostridium leptum* in the former is significantly higher than that of the latter [28]. Therefore, the increase of *Blautia* and decrease of *Clostridium sensu stricto* in the relative abundances in the MFMT group on day 7 suggest that MFMT may contribute to the resistance of piglets to disease, thus promoting colon health.

On day 21, our observation showed that AM had perturbed colonic microflora after being administered since birth (e.g., the richness estimator (Chao) was increased in the colon on day 21). This discovery was opposite to a previous study reporting that doses of parenteral amoxicillin administration affected the microbiota composition and diversity in neonatal piglets [29]. The inconsistent results could be due to the different animal model used and the different sampling conditions (e.g., gut segments and sampling time). The relative abundance of *Proteobacteria* in the ileum of the AM group was significantly reduced compared to the CO group, which is consistent with Dethlefsen’s study [30], but no significant change was found in the MFMT group. One potential speculation may be the long-lasting effects of antibiotics [31]. Meanwhile, the decrease of *Streptococcus* in the ileum and the increase of *Lachnoclostridium* in the colon also provide evidence for our conjecture. In addition, our findings indicated a weak direct effect of antibiotics on *Streptococcus*, which is in line with previous studies of Sullivan [32].

In observing the alteration in the GIT microbiota of piglets by MFMT, there was an interesting phenomenon that the effect of MFMT on the colon was greater than that on the stomach and the small intestine. Geng et al. transplanted porcine fecal microbiota suspension to neonatal piglets, and deep sequencing revealed significant changes in the diversity and compositions of the colon flora [33]. Zhang et al. also showed that FMT shifted the colon microbiota [34]. The anterior digestive tract microbiota were more susceptible to the flow of exogenous microbiota suspension. The reason for the contrary results to this theory requires further investigation. We suspect that it may be due to the influence of the fermentation product contained in the parental fecal bacteria on the colon of piglets.

### 3.2. Effects of Early Maternal Fecal Microbiota and Antibiotics Intervention on the Metabolite Profiles

Given the significant differences in colonic flora, we next investigated changes in metabolite profiles in the colon. In this study, both treatments (i.e., strengthen or weaken environmental factors) improved energy metabolism and amino acid (AA) metabolism on day 7. Compared to the CO group, galactose, starch, and sucrose metabolism were significantly changed in the two treatment groups, and sucrose associated with these pathways was also significantly decreased, indicating that changes in these metabolic pathways may be caused by sucrose. Sucrose can be decomposed into galactose and fructose, while galactose metabolism is the energy fuel [35]. Therefore, the reduction of sucrose content may enhance the metabolism of galactose and then facilitate the production of energy. Meanwhile, aspartate metabolism and protein biosynthesis-related pathways in both treatments also changed significantly compared to the CO group. Aspartate is the raw material of protein synthesis [36], and lysine, methionine, threonine, and isoleucine synthesis all use aspartate as a precursor. Thus, both treatments were beneficial to the protein synthesis in the hindgut of piglets. However, the two different microbial inoculation patterns had significant differences in carbohydrate and AA metabolism. Compared to the CO group, metabolic pathways (i.e., pyruvate metabolism, gluconeogenesis, and citric acid cycle), metabolites (i.e., fumaric acid and lactic acid)—all associated with the TCA cycle—increased in the MFMT group, and not in the AM group. The TCA cycle is closely related to the storage and utilization of energy [37] and is crucial to the health of piglets. Thus, changes in the TCA cycle suggest that adding maternal fecal microbiota may be more conducive to energy metabolism than antibiotics, but the mechanism still needs further investigation. Compared to the CO group, in the MFMT group asparagine increased, putrescine decreased, and metabolic pathways changed, including beta-alanine metabolism, methionine metabolism, lysine degradation, aspartate metabolism, tyrosine metabolism, phenylalanine and tyrosine metabolism, arginine and proline metabolism, urea cycle, and ammonia recycling. Putrescine is derived from the decarboxylation of ornithine [38], which produces adverse reactions in the host, such as genotoxicity [39]. The decrease in putrescine in the colon suggested that amino-acid decarboxylation reduction is beneficial to the luminal environment after MFMT [40]. Meanwhile, these increases in relation to AAs can increase protein synthesis [41]. AA metabolism is relative to carbohydrate metabolism, as the AA catabolism process contributes essential carbon skeleton intermediates to the TCA cycle and gluconeogenesis [42]. Therefore, these alternations indicate that early intervention with maternal bacteria may increase protein synthesis and enhance energy metabolism.

The present study found that MFMT increased the abundances of some bacteria associated with short-chain fatty acid (SCFA) metabolism in the colon, which may be a positive feedback on the enhancement of the TCA cycle. The abundances of OTU5 (member of *Bacteroidales S24-7 group*) and OTU41 (member of *Blautia*) in the MFMT group were increased in the colon (Appendix A). *Bacteroidales S24-7 group* and *Blautia* bacteria are reported to produce propionate and acetic acid that keep energy balance [27,43] and benefit colon health. The shifts in the SCFA-related bacteria and the metabolic pathways may originate from the maternal fecal microbiota or the interaction between maternal microbiota and piglets’ colon.

On day 21, results showed a better performance of carbohydrate metabolism and AA metabolism in the AM group than those in the MFMT group, thus illustrating long-lasting metabolic effects after antibiotic administration, which is line with previous studies [44,45]. At this time, the effects on the gastrointestinal microbiota and metabolite profiles of MFMT were relatively faded. Alternations in the gut microbiota and catabolism of nitrogen compounds and concentrations of AAs were observed on day 120 after postnatally adding antibiotics in a previous study [46]. A possible explanation is that piglets had restored what was previously destroyed [47], indirectly proving the security of the maternal fecal microbiota suspension. This also provides a reference for the future dosage and feeding length of the suspension.

## 4. Materials and Methods

### 4.1. Ethics Statement

The experiment was approved and conducted under the supervision of the Animal Care and Use Committee of Nanjing Agricultural University (Nanjing, Jiangsu province, China) (SYXK2017-0007). All pigs were raised and maintained on a local commercial farm under the care of the Animal Care and Use Guidelines of Nanjing Agricultural University.

### 4.2. Preparation of Maternal Fecal Microbiota Suspension

The preparation of the suspension followed the procedure of a previous study [48]. In brief, approximately 50 g fresh anaerobically maternal feces from a candidate pregnant sow was mixed with 250 mL sterile 0.9% NaCl solution. The mixture was homogenized and filtered with sterile gauze. The solution was centrifuged at 2000 rpm for 10 min, then the supernatant was transferred to 10 mL freezing tubes that were filled with CO_2_. The whole procedure was conducted in anaerobic conditions.

### 4.3. Experimental Design

In our study, five litters of healthy neonatal piglets (Duroc × Landrace × Yorkshire, nine piglets in each litter) were used. Each litter was randomly assigned into the CO, AM, or MFMT groups, with three piglets in each group. On days 1–6, the piglets in the AM group and the CO group were orally administrated once a day with the same volume of amoxicillin liquid (20 mg/kg) and physiological saline (0.9% NaCl), respectively, while 3 mL maternal fecal microbiota solution (>10^9^ CFU/mL) was offered to the piglets in the MFMT group. All pigs had access to breast milk and water ad libitum and had no other creep feed during the experiment.

### 4.4. Sampling

At days 7 and 21, five piglets (*n* = 5) from each group were randomly selected (one piglet from each litter) and slaughtered. The stomach, ileum, and colon digesta were collected and preserved at −20 °C for bacterial analysis. In addition, the collection of digesta from the proximal colon was stored in liquid nitrogen until metabonomic detection.

### 4.5. Illumina MiSeq Sequencing and Bioinformatics Analysis

Total genomic DNA in the stomach, ileum, and colon digesta were extracted with an accessible DNA extraction kit (Qiagen, Hilden, Germany) following the manual of the manufacturer. The concentration of DNA was quantified using a Nanodrop spectrophotometer (Thermo, Wilmington, DE, USA). Universal primers 515F and 907R were used for amplifying the V4-V5 region of the bacterial 16S rRNA gene in accordance with previously mentioned methods [49]. The purified amplicons were pooled in equimolar and paired-end sequenced (2 × 250) on an Illumina MiSeq platform according to the standard protocols. Raw Illumina fastq files were trimmed, filtered, and analyzed with the Mothur program (version 1.32.0) as per the description of a previous study [50]. Through quality controlling, 4,242,267 sequences from all 90 samples were selected, with the average length of 423.41 bp.

### 4.6. Real-Time PCR Quantification of Total Bacteria

Total bacteria were quantified by using primer set Bact1369/Prok1492 on an Applied Biosystems 7300 Real-Time PCR System (Applied Biosystems, USA) with SYBR fluorescence dye [51]. The PCR was fulfilled following the methods of a previous study [52].

### 4.7. Sample Preparation and GC-MS Analysis

GC-MS analysis was performed following the methods of a previous study [53]. In brief, weighed colonic digesta was mixed with water (1:3), centrifuged, and 50 µL supernatant was blended with 200 µL of methanol (involving [^13^C_2_]-myristic acid, 12.5 µg/mL). After 1 h of incubation, 100 µL supernatant was evaporated with a SpeedVac concentrator (Savant Instruments, Framingdale, NY, USA). The dried material was methoximated and trimethylsilylated and was then ready for analyzing metabolites using the GC–MS system, as previously mentioned [54].

### 4.8. GC-MS Data Acquisition and Processing

Metabolites were identified by contrasting the databases of the National Institute of Standards and Technology Library 2.0 and Wiley 9. Peak area was normalized to [^13^C_2_]-myristic acid before any subsequent process. SIMCA-P 13.0 (Umetrics, Umea, Sweden) was used for multivariate analysis. The preprocessed GC-MS data were handled with OPLS–DA. The compounds with *p*-value < 0.1 and VIP value > 1.0 were chosen as discriminated metabolites. Metaboanalyst (version 3.0) was used to identify the metabolic pathways and metabolite set enrichment analysis [55].

### 4.9. Statistical Analysis

The statistical analysis was conducted as a randomized complete design with SPSS (version 20). One litter was considered as one experimental unit (i.e., *n* = 5). The data were analyzed by one-way ANOVA to evaluate differences among treatments on each day, and *p* < 0.05 was regarded as significantly different.

## 5. Conclusions

The present study showed that MFMT and AM discriminatively modulated gastrointestinal microflora and altered the metabolic profiles of piglets. The early intervention of MFMT or AM showed that MFMT had beneficial effects on GIT microbiota and the metabolic profiles of piglets on day 7, while they were scarcely affected on day 21, but the effects of AM seemed to persist. A location-specific change resulting from MFMT found in our study deserves further investigation.

## Figures and Tables

**Figure 1 metabolites-08-00089-f001:**
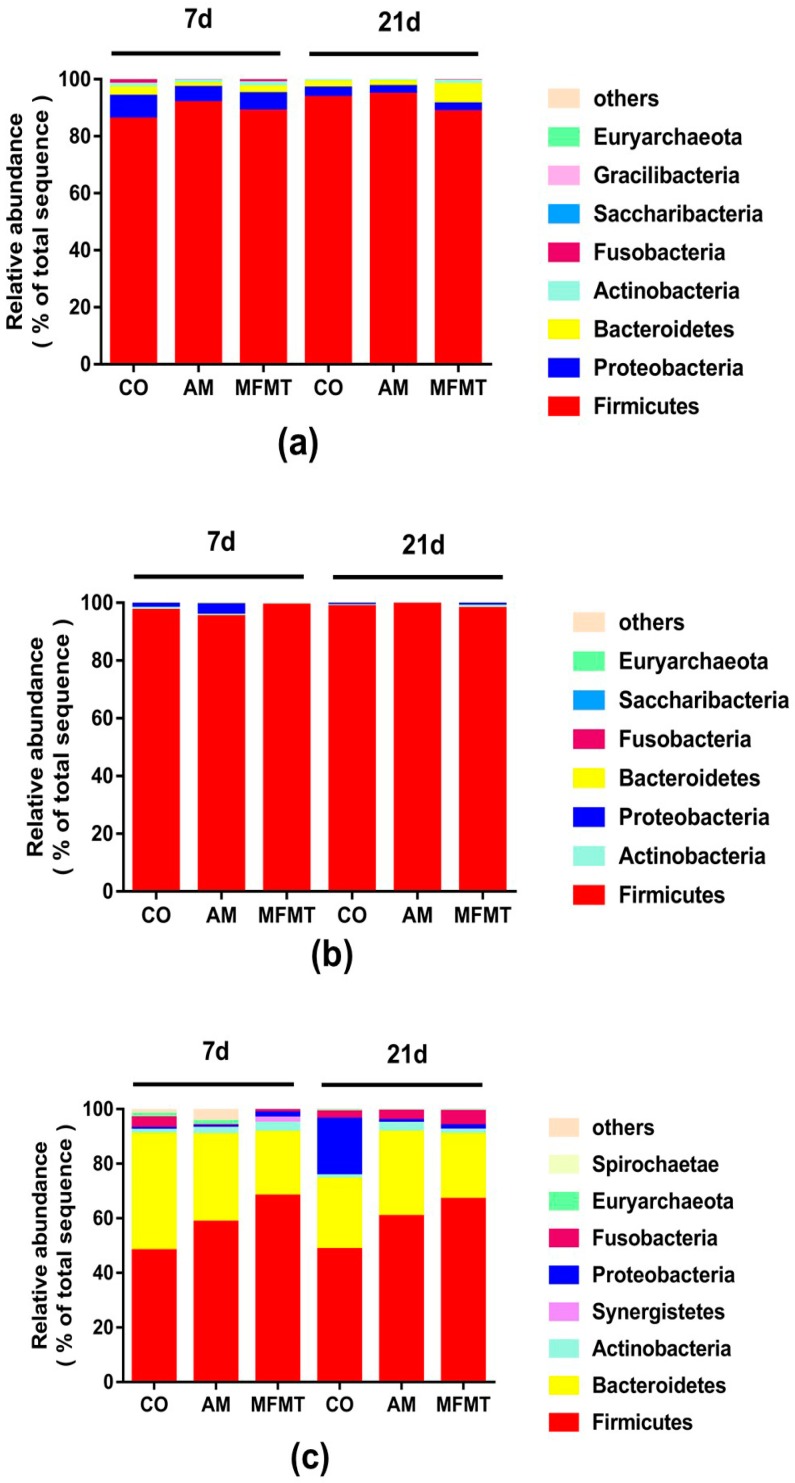
Relative abundances of the microbial phylum in the stomach (**a**), ileum (**b**), and colon (**c**) of piglets in the maternal fecal microbiota transplantation (MFMT), amoxicillin (AM), and control (CO) groups.

**Figure 2 metabolites-08-00089-f002:**
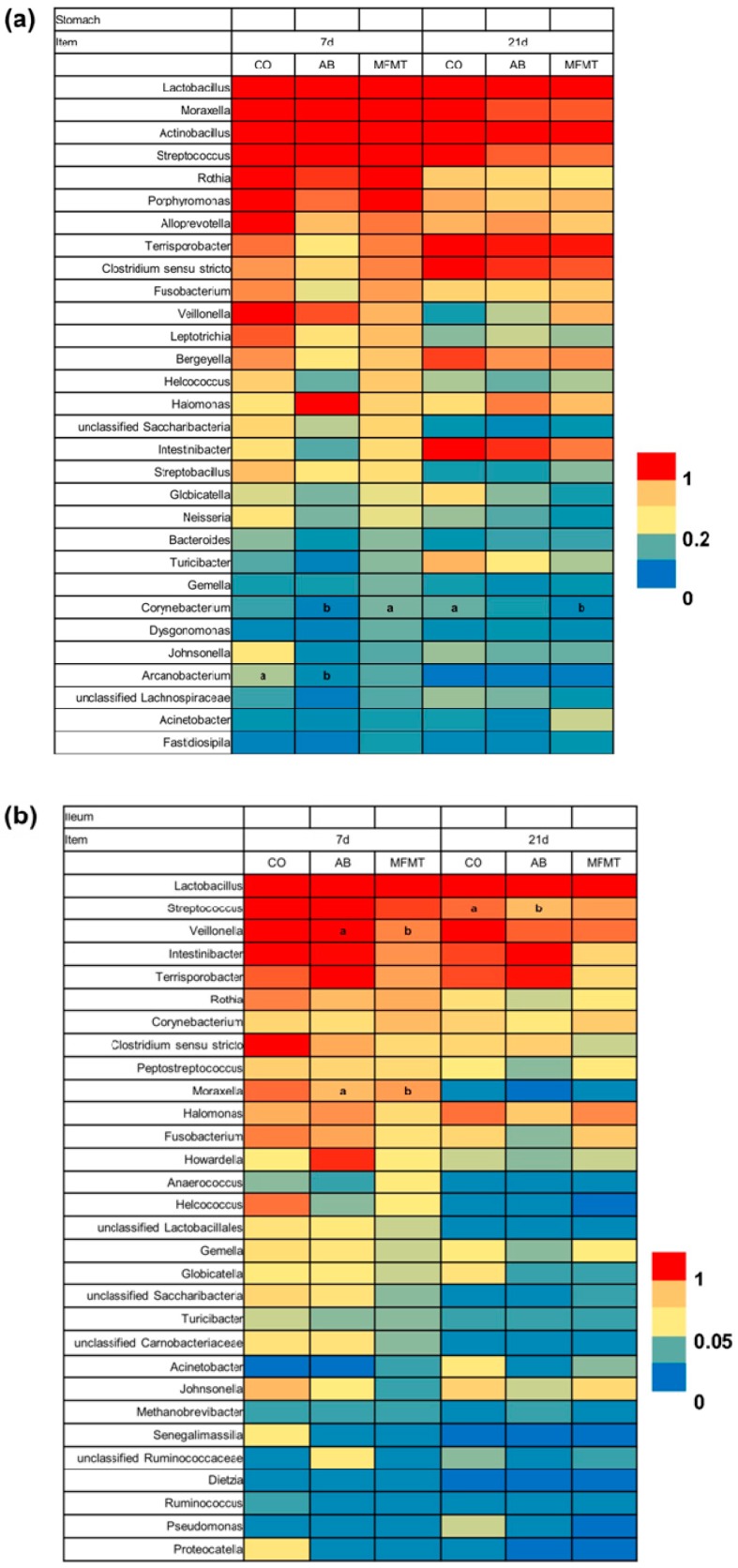
Relative abundances of the microbial genus in the stomach (**a**), ileum (**b**), and colon (**c**) of piglets in the maternal fecal microbiota transplantation (MFMT), amoxicillin (AM), and control (CO) groups. Mean values within a line with different superscript letters (a, b) on each day differ significantly (*p* < 0.05).

**Figure 3 metabolites-08-00089-f003:**
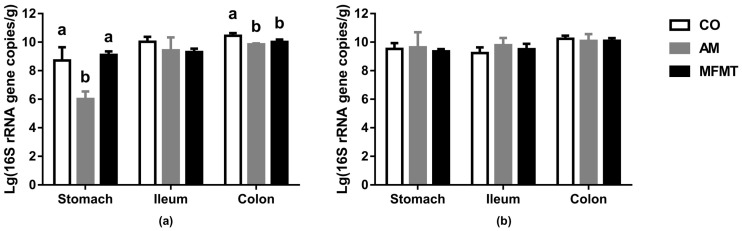
Total bacteria number in the stomach, ileum, and colon of piglets in the maternal fecal microbiota transplantation (MFMT), amoxicillin (AM), and control (CO) groups on days 7 (**a**) and 21 (**b**). Mean values of each gastrointestinal segment with different superscript letters (a, b) differ significantly (*p* < 0.05).

**Figure 4 metabolites-08-00089-f004:**
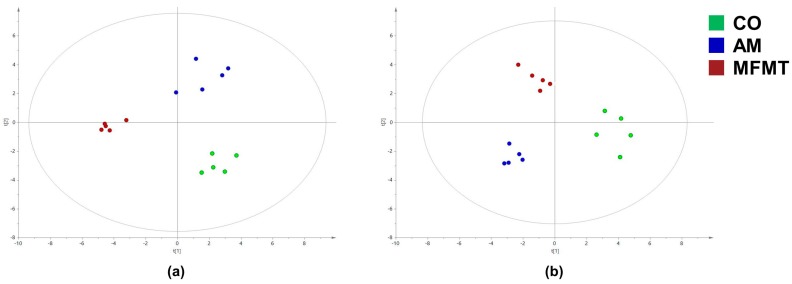
Orthogonal partial least squares discriminant analysis (OPLS–DA) of the microbial metabolites in the colonic contents from pigs in the maternal fecal microbiota transplantation (MFMT), amoxicillin (AM), and control (CO) groups on days 7 (**a**) and 21 (**b**).

**Figure 5 metabolites-08-00089-f005:**
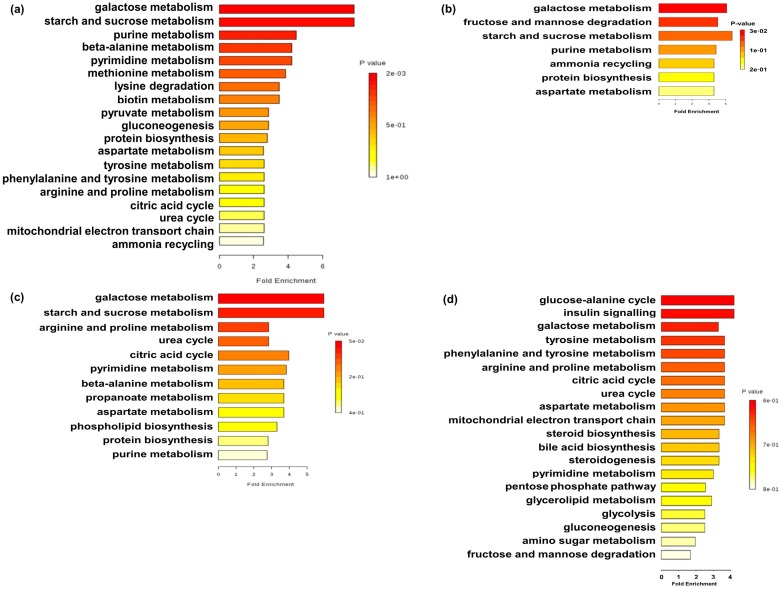
Metabolome view map of the common metabolites (variable importance projection >1.00) identified in the colon in the maternal fecal microbiota transplantation (MFMT), amoxicillin (AM), and control (CO) groups on days 7 (**a**,**b**) and 21 (**c**,**d**). The rectangle color is based on the *p*-value, and the rectangle length is determined from the pathway impact values. Longer lengths and darker colors represent greater pathway enrichment and greater pathway impact values, respectively. (**a**,**c**) MFMT vs. CO; (**b**,**d**): AM vs. CO.

**Table 1 metabolites-08-00089-t001:** Diversity estimation of the 16S rRNA gene libraries from microbiota in the stomach, ileum, and colon of piglets in the maternal fecal microbiota transplantation (MFMT), amoxicillin (AM), and control (CO) groups (*n* = 5).

Item	7 Days			21 Days		
	CO	AM	MFMT	CO	AM	MFMT
Stomach						
ACE	312.12 ± 28.82 ^b^	259.54 ± 16.80 ^b^	412.82 ± 30.39 ^a^	428.91 ± 26.10	413.41 ± 47.68	344.95 ± 34.48
Chao	287.15 ± 24.33	241.77 ± 17.79 ^b^	352.51 ± 30.25 ^a^	410.98 ± 23.96	390.91 ± 28.42	339.19 ± 42.13
Shannon	2.38 ± 0.35	1.93 ± 0.18	2.36 ± 0.06	2.21 ± 0.22	1.92 ± 0.04	2.33 ± 0.10
Simpson	0.21 ± 0.05	0.28 ± 0.04	0.22 ± 0.02	0.23 ± 0.03	0.31 ± 0.03 ^a^	0.21 ± 0.02 ^b^
Ileum						
ACE	226.27 ± 39.13	240.86 ± 24.35	247.66 ± 30.30	255.18 ± 10.87	222.05 ± 26.10	234.52 ± 29.94
Chao	194.19 ± 28.53	218.32 ± 27.09	201.50 ± 14.65	241.54 ± 11.05	200.68 ± 25.40	214.92 ± 30.62
Shannon	2.19 ± 0.25	2.28 ± 0.26	2.10 ± 0.07	1.92 ± 0.05	1.80 ± 0.07	2.06 ± 0.18
Simpson	0.20 ± 0.04	0.20 ± 0.05	0.21 ± 0.01	0.28 ± 0.02	0.28 ± 0.03	0.25 ± 0.06
Colon						
ACE	363.86 ± 27.30	349.31 ± 32.34	372.66 ± 27.10	385.5 ± 88.51	545.61 ± 28.07	466.36 ± 22.04
Chao	348.04 ± 18.91	341.52 ± 29.45	379.41 ± 32.03	385.23 ± 88.94 ^b^	563.07 ± 33.34 ^a^	472.59 ± 20.42
Shannon	3.58 ± 0.07	3.63 ± 0.13	3.38 ± 0.18	3.20 ± 0.72	4.03 ± 0.13	3.88 ± 0.19
Simpson	0.06 ± 0.01	0.05 ± 0.01	0.09 ± 0.02	0.21 ± 0.16	0.04 ± 0.01	0.06 ± 0.02

^a, b^ Mean values within a line with different superscript letters on each day differ significantly (*p* < 0.05).

**Table 2 metabolites-08-00089-t002:** Candidate colon compounds that differed in the maternal fecal microbiota transplantation (MFMT), amoxicillin (AM), and control (CO) groups on days 7 and 21.

Item	Metabolite	Biological Roles	Metabolic Subpathway	FC ^1^	*p* ^2^	VIP ^3^	FDR ^4^
**Day 7**							
MFMT vs. CO							
Carbohydrates	Sucrose	Disaccharides	Galactose metabolism	0.38	0.008	2.44	0.001
Others	1-Monohexadecanoylglycerol	Others	Others	0.72	0.032	1.84	0.005
Amino acids	Asparagine	Amino acids	Alanine, aspartate, and glutamate metabolism	4.15	0.032	1.05	0.007
Alkaloids	Hypoxanthine	Purine alkaloids	Purine metabolism	2.36	0.032	1.64	0.009
Amino acids	Lysine	Amino acids	Lysine metabolism	0.58	0.032	1.84	0.011
Others	1,3-Di-*tert*-butylbenzene	Others	Others	0.67	0.056	1.81	0.024
Lipids	Eicosanoic acid	Saturated fatty acids	Biosynthesis of unsaturated fatty acids	0.50	0.056	1.77	0.028
Lipids	Heptanoic acid	Straight chain fatty acids	Others	0.66	0.056	2.06	0.032
Nucleic acids	Uracil	Pyrimidines	Pantothenate and CoA biosynthesis	1.77	0.056	1.94	0.036
Others	2,4,6-Tri-*tert*-butylbenzenethiol	Others	Others	0.74	0.095	1.66	0.068
Carboxylic acid	Fumaric acid	Others	TCA cycle	1.44	0.095	1.42	0.075
Organic acids	Lactic acid	Hydroxycarboxylic acids	Glycolysis/gluconeogenesis	4.92	0.095	1.04	0.081
Alkaloids	Pipecolic acid	Piperidine alkaloids	Lysine degradation	0.31	0.095	1.45	0.088
Peptides	Putrescine	Amines	Arginine and proline metabolism	0.64	0.095	1.80	0.095
AM vs. CO							
Carbohydrates	Sucrose	Disaccharides	Galactose metabolism	0.57	0.008	2.49	0.001
Carbohydrates	Glucaric acid	Carbohydrates	Ascorbate and aldarate metabolism	2.35	0.056	1.79	0.016
Organic acids	Oxalic acid	Dicarboxylic acids	Glyoxylate and dicarboxylate metabolism	0.72	0.056	1.99	0.024
Carbohydrates	Sorbitol	Sugar alcohols	Polyol metabolism	1.90	0.056	2.11	0.032
Amino acids	Asparagine	Amino acids	Alanine, aspartate, and glutamate metabolism	1.87	0.095	2.01	0.068
Carbohydrate	Fructose	Ketoses	Fructose and mannose degradation	1.78	0.095	1.73	0.081
Alkaloids	Hypoxanthine	Purine alkaloids	Purine metabolism	1.71	0.095	1.72	0.095
**Day 21**							
MFMT vs. CO							
Carbohydrates	Sucrose	Disaccharides	Galactose metabolism	0.56	0.008	2.33	0.001
Nucleic acids	Uridine	Nucleosides	Pyrimidine metabolism	1.38	0.008	2.38	0.002
Others	2-Hydroxyglutaric acid	Others	Others	2.77	0.016	1.80	0.005
Organic acids	Citric acid	Tricarboxylic acid	TCA cycle	0.34	0.032	1.72	0.013
Peptides	Ethanolamine	Biogenic amines	Glycerophospholipid metabolism	2.28	0.056	1.55	0.028
Amino acids	Arginine	Amino acids	Arginine biosynthesis	1.54	0.095	1.68	0.057
Amino acids	Beta-alanine	Amino acids	Beta-alanine metabolism	2.48	0.095	1.86	0.067
Nucleic acids	Guanine	Purines	Purine metabolism	1.54	0.095	1.77	0.076
Amino acids	Ornithine	Other amino acids	Arginine biosynthesis	0.53	0.095	1.73	0.086
Amino acids	Urea	Amino acids metabolism relatives	Ornithine cycle	1.29	0.095	1.74	0.095
AM vs. CO							
Others	Sorbitol-6-phosphate	Others	Fructose and mannose metabolism	1.83	0.008	2.29	0.001
Nucleic acids	Uridine	Nucleosides	Pyrimidine metabolism	1.71	0.032	1.38	0.006
Steroids	Cholesterol	Cholestane derivatives	Steroid biosynthesis	0.47	0.056	1.53	0.015
Carboxylic acid	Fumaric acid	Others	TCA cycle	1.65	0.056	1.87	0.020
Carbohydrates	Glucose	Aldoses	Glycolysis/gluconeogenesis	1.78	0.056	1.94	0.025
Carbohydrates	Glycerol	Sugar alcohols	Glycerolipid metabolism	0.39	0.056	1.40	0.031
Carbohydrate	Ribose	Aldoses	Pentose phosphate pathway	2.08	0.056	1.91	0.036
Carbohydrates	Xylose	Aldoses	Pentose and glucuronate interconversions	1.57	0.056	1.58	0.041
Carbohydrates	Fructose-6-phosphate	Ketose	Glycolysis/gluconeogenesis	2.01	0.095	1.78	0.078
Others	Galactose-6-phosphate	Others	Others	1.23	0.095	1.01	0.086
Others	Mannose-6-phosphate	Others	Fructose and mannose metabolism	1.51	0.095	1.36	0.095

^1^ FC fold-change for the relative concentrations of metabolites; ^2^
*p*-value was less than 0.1; ^3^ VIP variable importance projection was obtained from the OPLS–DA model with a threshold of 1. ^4^ FDR false discovery rate.

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
