# Peer review of "Effects of Early Intervention with Maternal Fecal Microbiota and Antibiotics on the Gut Microbiota and Metabolite Profiles of Piglets"

_metabolites, 2018, doi:10.3390/metabo8040089_

Reviewer 1 Report

The manuscript “Effects of early intervention with maternal fecal microbiota and antibiotics on the gut microbiota and metabolite profiles of piglets” by Chunhui Lin, Jiajia Wan, Yong Su, Weiyun Zhu needs a major revision in order to be acceptable for publishing.

My remarks on the current manuscript include:

The text and style needs extensive editing to make the manuscript more understandable.  I would suggest a comprehensive proofreading by a native English speaker.

Line 70: First sentence of Results should be moved to Materials and methods.

Line 119 and 129: “MFMT affected”… Should this be “AM affected” instead, as the sentences compare AM to CO?

The statistical significance explained in Figure and Table captions is not easy to understand. First, the term “unlike superscript” should be changed to i.e. “different superscript letter”. I presume “a” is statistically different from “b”, but what about “ab”… is it different from both “a” and “b”? This needs to be explained better throughout the captions.

The Results section explains only a part of statistical differences presented in Figures and Tables, but not all. Example: results covering Table 1 explain increase of ACE for MFMT vs CO (stomach, day 7) and increase in Chao for AM vs CO (colon, day 21), but not the differences between groups in Chao (stomach, day 7) or Simpson (stomach, day 21). Why is not the complete statistic difference listed in Figures and Tables explained in the text?

Table 2 and 3 can be merged to a single Table. Table formatting should be performed so the Tables are uniform and easier to navigate.

Lines 138-146 in Discussion section are better suited for the Conclusion section, as they provide a recap of the study results.

Line 152: The role of Parabacteriodes as a potential pathogen is supported by an article from 1995. This is an old article not readily available on the internet and I couldn’t read it to confirm the author’s claims. However, more recent literature (Shahi et al, Gut Microbes. 2017; 8(6): 607–615. Or Cekenaviciute et al, PNAS October 3, 2017 114 (40) 10713-10718) reports of protective role of Parabacteriodes, degrading and fermenting complex polysaccharides, inducing SCFA production in MS, attenuating colitis in mice (Kverka et al, Clin. Exp. Immunol. 2011; 163: 250-259.). The effect MFMT has on Blautia (colon, day 7) was not explained at all.

I’m puzzled by Chao index increase by AM (colon, day 21). This is not supported by the article the authors are citing (26), as the article cited claims “antibiotic   administration   caused   a   shift   in   gut bacterial population towards decrease of the diversity and richness of the bacterial community”, which is complete opposite of what the authors are suggesting.

The complete Discussion section is not acceptable in this format and needs a substantial rework. The authors do not explain their results well, some of the cited articles do not support the claims made in the text.

Author Response

Responses to Reviewer:

We thank the reviewer’s comment. These suggestions are very important for us to better improve the text. We have carefully checked and corrected the entire manuscript into the revised manuscript.

Comment 1: Line 70: First sentence of Results should be moved to Materials and methods.

Response: We have now moved the sentence “Through the quality controlling, 4,242,267 sequences from all 90 samples were selected, with the average length was 423.41 bp.” from Results to Materials and Methods in the revision.

Comment 2: Line 119 and 129: “MFMT affected”… Should this be “AM affected” instead, as the sentences compare AM to CO?

Response: The word “MFMT”should be replace with “AM”. We now corrected the mistake.

Comment 3: The statistical significance explained in Figure and Table captions is not easy to understand. First, the term “unlike superscript” should be changed to i.e. “different superscript letter”. I presume “a” is statistically different from “b”, but what about “ab”… is it different from both “a” and “b”? This needs to be explained better throughout the captions.

Response: Thanks for the comment. We have now changed “The mean values within a column with unlike superscript letters are significantly different (P < 0.05)” to “mean values with different superscript letters differ significantly (P < 0.05)” in Table 1, Figure 2 and Figure 3 captions in the revision. Additionally, the letters “ab” in Figure and Table captions meant there was no significant difference from “a” or “b” (P > 0.05).

Comment 4: The Results section explains only a part of statistical differences presented in Figures and Tables, but not all. Example: results covering Table 1 explain increase of ACE for MFMT vs CO (stomach, day 7) and increase in Chao for AM vs CO (colon, day 21), but not the differences between groups in Chao (stomach, day 7) or Simpson (stomach, day 21). Why is not the complete statistic difference listed in Figures and Tables explained in the text?

Response: Thanks for the comment. We have now added the information in the revision as “The Chao value in the MFMT group was greater (P < 0.05) than that in the amoxicillin treatment (AM) group. On day 21, the diversity indices Simpson in the MFMT group significantly decreased (P < 0.05) in comparison with the AM group.”

In addition, we have now added “genera Corynebacterium was significantly increased (P < 0.05) in relative abundance by MFMT as compared to the AM group on day 7, and the reverse occurred (P < 0.05) in comparison with the CO group on day 21.” in the revision.

And we have now added “In the ileum, the abundance of Veillonella and Moraxella in the MFMT group significantly declined (P < 0.05) in comparison with the AM group on day 7 (Figure 2b).” (Lines 84-85) in the revision.

Comment 5: Table 2 and 3 can be merged to a single Table. Table formatting should be performed so the Tables are uniform and easier to navigate.

Response: Thanks for the valuable comment. We have now merged Tables 2 and 3 to a single Table 2 (Lines 485-489) and uniformed the tables in the revision.

Comment 6: Lines 138-146 in Discussion section are better suited for the Conclusion section, as they provide a recap of the study results.

Response: As Lines 138-146 in Discussion section was already expressed in the primary Conclusion section. So, we have now deleted this part. And we also emended “The current experiment aimed to study short- and long-term effects of early intervening with the maternal fecal microbiota and antibiotics on the gastrointestinal tract (GIT) microbial composition and metabolite profiles of neonatal piglets.” to “For the first time, as we have seen, the current experiment combined the high-throughput sequencing and microbiome analysis to study short- and long-term effects of early intervening with the maternal fecal microbiota and antibiotics on the gastrointestinal tract (GIT) microbial composition and metabolite profiles of neonatal piglets.” in the revision.

Comment 7: Line 152: The role of Parabacteroides as a potential pathogen is supported by an article from 1995. This is an old article not readily available on the internet and I couldn’t read it to confirm the author’s claims. However, more recent literature (Shahi et al, Gut Microbes. 2017; 8(6): 607–615. Or Cekenaviciute et al, PNAS October 3, 2017 114 (40) 10713-10718) reports of protective role of Parabacteroides, degrading and fermenting complex polysaccharides, inducing SCFA production in MS, attenuating colitis in mice (Kverka et al, Clin. Exp. Immunol. 2011; 163: 250-259.). The effect MFMT has on Blautia (colon, day 7) was not explained at all.

Response: Thanks for the reviewer’s comment. We are very sorry that these reports on Parabacteroides have not been retrieved. After reading these literature, we believe that there is no certain conclusion on the role of Parabacteroides. A study also revealed that Parabacteroides involved in infectious diseases [1]. To clarify, we have now revised “Some researchers found that Parabacteroides was responsible for infectious diseases, mainly bacteremia [24]. Therefore, the decrease in Parabacteroides in the colon on day 7 in our study showed a potential role of MFMT in strengthening intestinal health. However, recent studies demonstrated some Parabacteroides species could protectd colon and attenuated colitis [25, 26], however, further study is needed for understanding the mechanism for reducing Parabacteroides by MFMT.” (Lines 153-155) in the revision.

In addition, we have now added the explanation about the effect of MFMT on Blautia. The explanation is “It had been reported that Blautia produced acetate and succinate provided as important energy sources that conductive to colonic health.” in the revision.

Furthermore, to ensure consistency, we have now changed “Therefore, reduced in the relative abundance of Parabacteroides and Clostridium sensu stricto in the MFMT group suggesting MFMT may contribute to the resistance of piglets to disease, thus promoting the colon health.” to “Therefore, increased in the relative abundance of Blautia and reduced in Clostridium sensu stricto in the MFMT group on day 7 suggesting MFMT may contribute to the resistance of piglets to disease, thus promoting the colon health.” in the revision.

Reference cited here is as follows:

[1] Boente, R.F.; Ferreira, L.Q.; Falcao, L.S.; Miranda, K.R.; Guimaraes, P.L.S.; Santos-Filho, J.; Vieira, J.M.B.D.; Barroso, D.E.; Emond, J.-P.; Ferreira, E.O., et al. Detection of resistance genes and susceptibility patterns in Bacteroides and Parabacteroides strains. Anaerobe 2010, 16, 190-194; DOI:10.1016/j.anaerobe.2010.02.003.

Comment 8: I’m puzzled by Chao index increase by AM (colon, day 21). This is not supported by the article the authors are citing (26), as the article cited claims “a antibiotic administration caused a shift in gut bacterial population towards decrease of the diversity and richness of the bacterial community”, which is complete opposite of what the authors are suggesting.

Response: Thanks for the comment. Discussion was added in the revised vertion as ” On day 21, our observation showed that AM had perturbation of colonic microflora after administered since birth, such as the richness estimator (Chao) was increased in the colon on day 21. This discovery was opposite to a previous study reporting that doses of parenteral amoxicillin administration affected the microbiota composition and diversity in neonatal piglets [28]. The inconsistent results could be due to the different animal model used and the different sampling conditions such as gut segments and sampling time.”

Reviewer 2 Report

The manuscript 393409 entitled "Effects of early intervention with maternal fecal microbiota and antibiotics on the gut microbiota and  metabolite profiles of piglets" by Lin et al have demonstrated that maternal fecal microbiota transplantation modulate gastrointestinal microbiota and alter the colonic metabolic profiles of piglets (long-term with a location-specific influence on the gastrointestinal microbiota). The research is carried out with excellente decription of all methodological steps. The study focus in microbiota is a current research hot topic and this investigation support that. Finally, the conclusion is supported by the results and material and method described. It is a good piece of experimental research.

Some minimal changes are required in the names of phylum.

Author Response

Comment: Some minimal changes are required in the names of phylum.

Response: The names of phylum were italicized.

Round  2

Reviewer 1 Report

Thank you for implementing the changes requested. I still think the manuscript needs extensive English proofreading in both language and style. Perhaps the authors can forward the manuscript to some of their English speaking collaborators for editing?

As for the additional comments:

Comment 3: The statistical significance explained in Figure and Table captions is not easy to understand. First, the term “unlike superscript” should be changed to i.e. “different superscript letter”. I presume “a” is statistically different from “b”, but what about “ab”… is it different from both “a” and “b”? This needs to be explained better throughout the captions.

Response: Thanks for the comment. We have now changed “The mean values within a column with unlike superscript letters are significantly different (P < 0.05)” to “mean values with different superscript letters differ significantly (P < 0.05)” in Table 1, Figure 2 and Figure 3 captions in the revision. Additionally, the letters “ab” in Figure and Table captions meant there was no significant difference from “a” or “b” (P > 0.05). Figures and tables:

Additional comment: The changes were not implemented for tables in supplementary information (Table S1, S2 and S3). Additionally, if “ab” is not statistically different from either “a” or “b” then it is not necessary to include “ab” label at all, as it makes a confusion. I would suggest completely removing “ab” label from Tables and Figures.  

Author Response

Thank you very much for your careful comments and suggestions on our paper (Manuscript ID: 393409). Your comments are gratefully received. With your guidance, we have now checked the manuscript and revised it. We appreciate very much for your patience and valuable comments and please let me know your further advice.

Thank you for implementing the changes requested. I still think the manuscript needs extensive English proofreading in both language and style. Perhaps the authors can forward the manuscript to some of their English speaking collaborators for editing?

Response: Thank you very much, in the new version English was checked carefully, mistakes on grammar were corrected as possible as we can.

Additional comment: The changes were not implemented for tables in supplementary information (Table S1, S2 and S3). Additionally, if “ab” is not statistically different from either “a” or “b” then it is not necessary to include “ab” label at all, as it makes a confusion. I would suggest completely removing “ab” label from Tables and Figures.

Response: In the revised version, tables in supplementary information were also corrected. “ab” label from Tables and Figures was also removed.